# Regression-based Deep-Learning predicts molecular biomarkers from pathology slides

Omar S. M. El Nahhas [1], Chiara M. L. Loeffler[1,2], Zunamys I. Carrero [1], Marko van Treeck[1], Fiona R. Kolbinger [1,3], Katherine J. Hewitt[1], Hannah S. Muti[1,3], Mara Graziani [4], Qinghe Zeng [5], Julien Calderaro[6], Nadina Ortiz-Brüchle[7,8], Tanwei Yuan[9], Michael Hoffmeister[9], Hermann Brenner [9,10,11], Alexander Brobeil[12,13], Jorge S. Reis-Filho [14] & Jakob Nikolas Kather [1,2,15,16] ✉

Deep Learning (DL) can predict biomarkers from cancer histopathology. Several clinically approved applications use this technology. Most approaches, however, predict categorical labels, whereas biomarkers are often continuous measurements. We hypothesize that regression-based DL outperforms classification-based DL. Therefore, we develop and evaluate a self-supervised attention-based weakly supervised regression method that predicts continuous biomarkers directly from 11,671 images of patients across nine cancer types. We test our method for multiple clinically and biologically relevant biomarkers: homologous recombination deficiency score, a clinically used pan-cancer biomarker, as well as markers of key biological processes in the tumor microenvironment. Using regression significantly enhances the accuracy of biomarker prediction, while also improving the predictions' correspondence to regions of known clinical relevance over classification. In a large cohort of colorectal cancer patients, regression-based prediction scores provide a higher prognostic value than classification-based scores. Our open-source regression approach offers a promising alternative for continuous biomarker analysis in computational pathology.

The collection and pathological examination of tissue specimens is used for accurate diagnosis of patients with malignant tumors, providing information related to histology grade, subtype, stage and other tumor biomarkers. Digital pathology describes the computational analysis of tissue specimen samples in the form of whole slide images (WSI). Numerous studies have shown that alterations in individual genes[1-3], microsatellite instability[4-6], and the expression of individual genes[7] or expression patterns of groups of genes[8,9] can be predicted directly from WSI. This research area has also enabled genetic changes to be correlated with morphological patterns (i.e. genotypic-phenotypic correlations)[10], which facilitates the prediction of patient outcome[11]. Consistent with their clinical application, several of these methods have been approved for clinical use by regulatory agencies[12], to the extent that the prediction of biomarkers from pathological

diagnostic workflows based on deep learning (DL) is becoming increasingly relevant, not only in the research setting, but also as a de facto clinical application[2,12,13].

The prediction of genotypic-phenotypic correlations, which involves predicting genetic biomarkers from WSIs, is a weakly supervised problem in DL. To accomplish this task, a DL model correlates phenotypic features from WSIs with a single ground truth obtained from molecular genetic sequencing of tumor tissue at the patient level. Nevertheless, as these WSI are of gigapixel resolution, neural network processing requires breaking them into smaller regions referred to as tiles or patches. These regions may, however, contain less relevant tissues such as connective tissue or fat, which might not contribute to biomarker predictability[14]. To address this issue, attention-based multiple instance learning (attMIL) is the predominant technical

approach that is currently used[15–18]. To implement this strategy, feature vectors are first extracted from pre-processed tiles. These vectors are then aggregated by a multi-layer perceptron with an attention component, allowing for a patient-level prediction of the WSI.

Despite the current attMIL approach yielding a high accuracy for biomarker prediction from WSIs[15,19,20], almost all published approaches are limited to classification problems with categorical values (e.g. presence or absence of a genetic alteration)[1–3,8,11,21,22]. Nonetheless, the ground truth of many biomarkers is available as continuous values, which are then binarized prior to being utilized as ground-truth for DL. This is true for whole-genome duplications, copy number alterations, homologous recombination deficiency (HRD), gene expression values, protein abundance, and many other measurements. Studies that pursue regression analysis of continuous values often opt for dichotomization or custom thresholds for categorization. For example, prior to modeling, Fu et al. utilized a LASSO approach for the classification of continuous chromosome data into three classes[10]. Schmauch et al. trained a regression model to predict continuous biomarkers and subsequently used percentile thresholds for the evaluation of the models through a categorical representation[7]. Chen et al. performed feature extraction using Cox regression with L1 regularization, after which the risk scores were dichotomized into binary categories to predict disease free survival[23].

However, binarization or dichotomization of these values results in information loss[24], which presumably limits the performance of DL systems predicting these biomarkers from pathology slides. Alternatively, a more suitable approach to classification in histopathological WSI analysis would be regression. Regression is a modeling approach used to investigate the relationship between variables[25], such as morphological features from a WSI, and continuous numerical values, such as genetic biomarkers. To date, there is a paucity of data exploring this approach. Several studies have explored different approaches for predicting gene expression levels and spatial gene expressions from WSIs. Huang et al. utilized contrastive learning combined with a linear regression model to predict differential gene expression levels[26]. Similarly, Dawood et al. employed ordinary least squares regression to predict spatial gene expressions from WSIs[27]. Moreover, Mondol et al. and Hoang et al. employed convolutional neural network regression to predict mRNA expression levels of various genes from pathology slides[28,29]. Schirris et al. utilized multiple instance learning regression to predict stromal tumor infiltrating lymphocytes directly from histopathology slides[30]. However, the study acknowledged the absence of an attention mechanism as a potential limitation, which could have contributed to improved accuracy in the predictions. The application of attention mechanisms in regression was investigated by Weitz et al. for predicting gene expressions from WSI, where a decrease in generalizability was observed in models with an attention component[31]. However, their analysis was limited by a small sample size in only a single cancer type. A recent study by Graziani et al. presented an approach to predict continuous values from pathological images using a form of attMIL[32], yet their regression network was not systematically compared and required more extensive validation with respect to the more-explored classification approach.

In this study, we systematically compared classification- and regression-based approaches for prediction of continuous biomarkers across multiple cancer types. We hypothesized that regression outperforms classification in weakly supervised analyses of pathology hematoxylin-and-eosin (H&E)-stained WSIs for biomarker predictability, the correspondence to regions of known clinical relevance and prognostic capability. In addition to various tumor entities, our work also explores several clinically relevant biomarkers represented as continuous numerical values. As a result, we developed a contrastively-clustered attention-based multiple instance learning (CAMIL) regression approach, which combines self-supervised learning (SSL) with attMIL, and systematically compared it to two state-of-the-art

approaches: the CAMIL classification approach, and the regression method proposed by Graziani et al.[32]. The comprehensive evaluation and application of regression versus classification methods across multiple datasets, organs, and biomarkers bridges a notable gap in the computational pathology literature.

## Results

### Regression predicts HRD from histology

We developed a regression-based DL approach which combines a feature extractor trained by SSL[33] and an attMIL[14] model (Fig. 1A, B), referred to as contrastively-clustered attention-based multiple instance learning (CAMIL) regression. We tested the abilities of this approach for prediction of HRD directly from pathology images. We chose HRD because it is a pan-cancer biomarker that is measured as a continuous score, but can be binarized at a clinically validated cutoff. We used the The Cancer Genome Atlas (TCGA) cohorts for breast cancer (BRCA), colorectal cancer (CRC), glioblastoma (GBM), lung adenocarcinoma (LUAD), lung squamous cell carcinoma (LUSC), pancreatic adenocarcinoma (PAAD), and endometrial cancer (UCEC) to train a regression DL model for each cancer type and evaluated their performance by cross-validation (Fig. 1C, D). To mitigate batch effects, which are problematic in the TCGA cohort, we used site-aware cross-validation splits[34]. We found that our CAMIL regression models were able to predict HRD status with AUROCs above 0.70 in 5 out of 7 tested cancer types. The area under the receiver operating characteristic (AUROC) with 95% confidence interval (CI) were 0.78 [0.75–0.81] in BRCA, 0.76 [0.65–0.87] in CRC, 0.64 [0.37–0.79] in GBM, 0.72 [0.62–0.81] in PAAD, 0.72 [0.67–0.77] in LUAD, 0.57 [0.52–0.63] in LUSC, and 0.82 [0.78–0.86] in UCEC (Fig. 2A, Supplementary Table 1). We validated the models on CPTAC, a set of external validation cohorts, in which images and HRD status were available for LUSC, LUAD, PAAD, UCEC (Fig. 2B). In these cohorts, the model achieved even higher AUROCs, reaching 0.68 [0.56–0.79] in PAAD, 0.81 [0.77–0.85] in LUAD, and 0.96 [0.93–0.98] in UCEC. The lowest AUROC was 0.62 [0.56–0.67] in LUSC (Supplementary Table 1). Together, these data show that regression-based DL can predict HRD status from pathology images alone.

### Regression outperforms the state-of-the-art classification-based approach

We compared the performance of our DL approach, CAMIL regression, against two state-of-the-art approaches: the Graziani et al. regression method[32] and the CAMIL classification method. In order to compare classification with regression, we chose the AUROC as an evaluation metric. In the site-aware-split test set of the TCGA cohort, CAMIL regression outperformed both of the previous approaches in HRD prediction for 5 out of the 7 tested cancer types, with GBM and LUSC exhibiting similar AUROCs (Fig. 2A, Supplementary Table 1). Significant performance differences were observed between CAMIL classification and Graziani et al. regression ($p \le 0.0167$) in the TCGA-BRCA cohort. A paired two-tailed DeLong's test revealed additional significant differences, in this case between CAMIL regression and Graziani et al. regression ($p \le 0.01$) in the TCGA-CRC cohort (Supplementary Table 2). In the external validation cohorts, no statistically significant differences are noted in AUROCs across the models (Supplementary Table 2). Of note, CAMIL regression exhibited lower variance in model performance across the 5-folds for most cancer types, as evidenced in both the internal (Fig. 2A) and external (Fig. 2B) cohorts. These findings suggest that CAMIL regression learns more robust features compared to CAMIL classification across different patient subsets. These data provide evidence that regression outperforms classification, even though the classification model was trained on curated binary categories using clinically-relevant cut-off points, and evaluated using the classification-specific AUROC metric.

Consequently, we investigated additional aspects of model performance which the AUROC does not capture[35]. For this, we compared

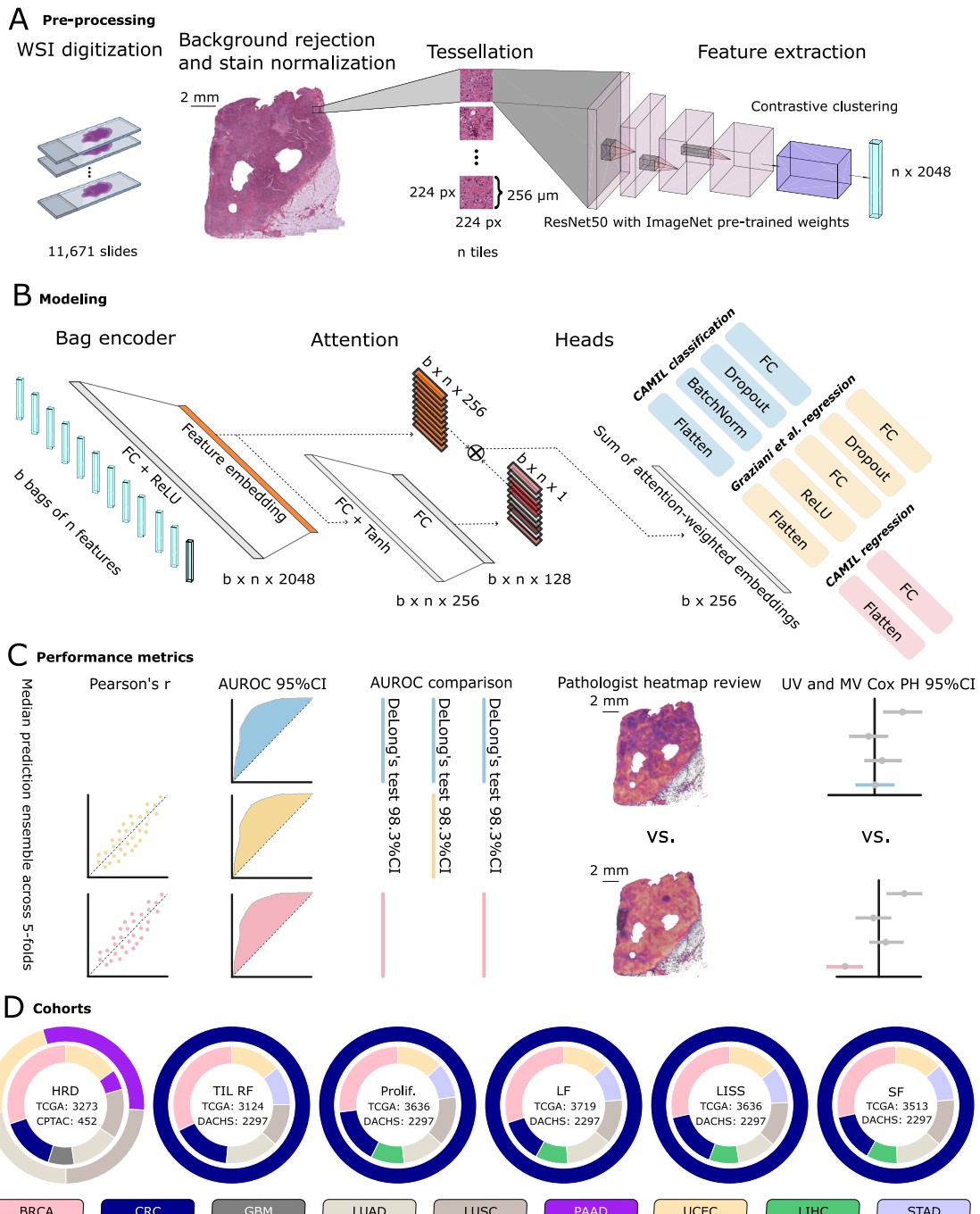

**Fig. 1 | End-to-end experimental workflow overview with image pre-processing, modeling, performance metrics and used cohorts. A** Image pre-processing pipeline and tile-level feature extraction by running inference on a ResNet50 with pre-trained ImageNet weights and retrieval contrastive clustering (RetCCL) model for a feature matrix for each patient. **B** Depiction of the modeling architecture utilizing attention-based multiple instance learning (attMIL) applied to the self-supervised extracted features. It incorporates three separately trained heads: one for CAMIL classification, one for regression following the method proposed by Graziani et al. and a third for the CAMIL regression method introduced in this study. **C** Performance metrics and their respective confidence intervals (CIs) used to assess the three separately trained heads of the model. Evaluation measures include Pearson's correlation coefficient (Pearson's r) for the regression models, and the Area Under the Receiver Operating Characteristic curve (AUROC) for all models. A paired two-tailed DeLong's test was conducted for the homologous recombination deficiency (HRD) and biological process biomarkers. Expert reviews of attention heatmaps were undertaken alongside univariable (UV) and multivariable (MV) Cox proportional-hazards (PH) models for the biological process models. **D** Chart representation of the cohorts used in this study, where the inner and outer circles denote which were utilized for training and external validation, respectively. Training cohorts are sourced from The Cancer Genome Atlas (TCGA) program for all clinical targets. External validation cohorts are derived from the Clinical Proteomic Tumor Analysis Consortium (CPTAC) effort and the Darmkrebs: Chancen der Verhütung durch Screening (DACHS) study, specifically for the HRD target and the biological process biomarkers, respectively. The biological process biomarkers considered include tumor infiltrating lymphocytes regional fraction (TIL RF), proliferation (Prolif.), leukocyte fraction (LF), lymphocytes infiltrating signature score (LISS), and stromal fraction (SF). The cancer types considered in this study are breast cancer (BRCA), colorectal cancer (CRC), glioblastoma (GBM), lung adenocarcinoma (LUAD), lung squamous cell cancer (LUSC), pancreas adenocarcinoma (PAAD), endometrial cancer (UCEC), liver hepatocellular carcinoma (LIHC), and stomach cancer (STAD). Source data are provided as a Source Data file. Slide icon adapted from "Icon Pack - Glass Slides", by BioRender.com (2023). Retrieved from https://app.biorender.com/biorender-templates.

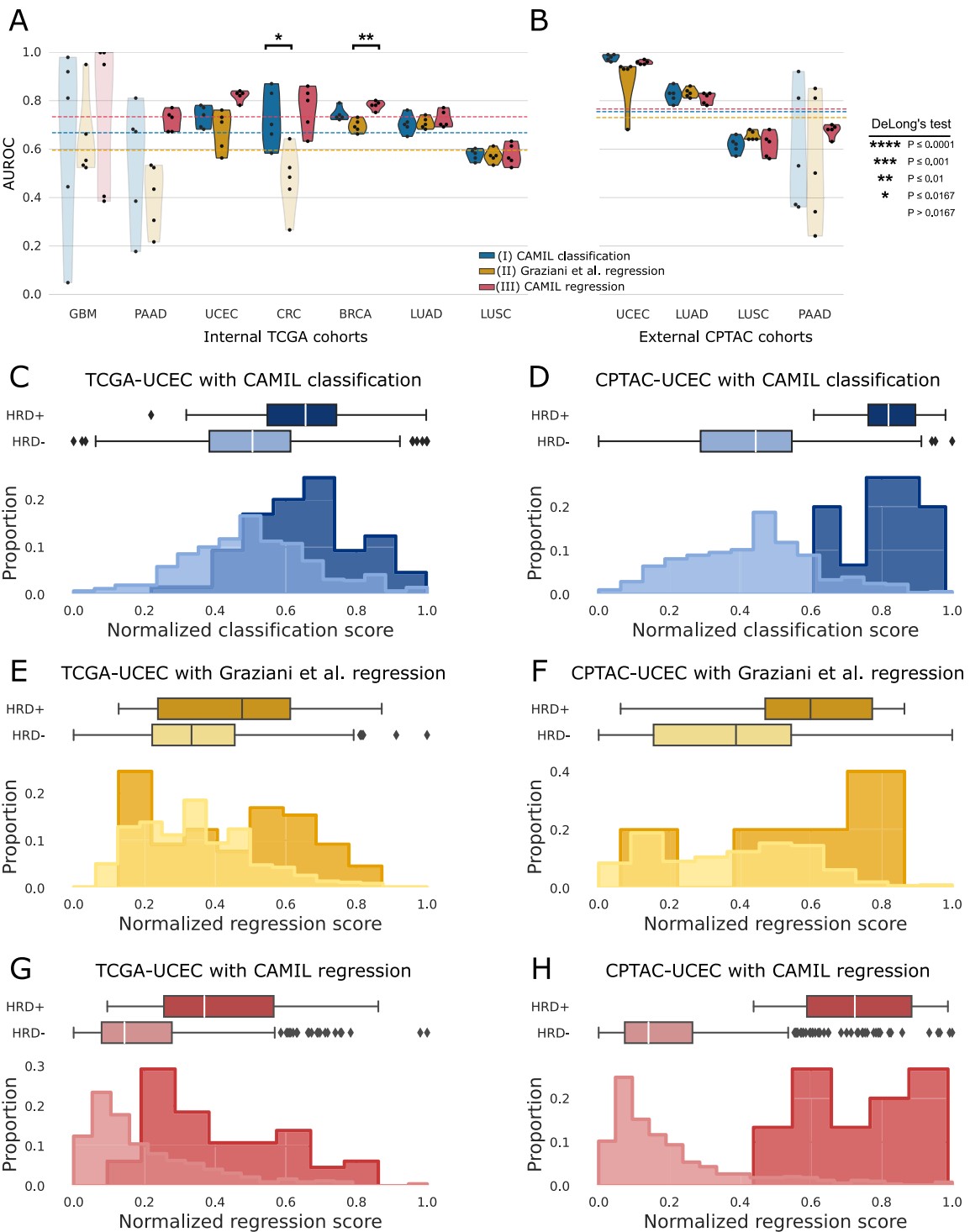

the three approaches by quantifying the absolute distance between the medians of the normalized scores for the positive and negative samples (Fig. 2C–H). For example, for detection of HRD status in endometrial cancer, the AUROC on the CPTAC test cohort was 0.98 [0.96–0.99] for CAMIL classification, 0.89 [0.79–0.98] for regression from Graziani et al. and 0.96 [0.93–0.98] for CAMIL regression. These differences were not statistically significant (Fig. 2B, Supplementary Table 2). When visualizing the distribution of HRD predictions from the models, both the CAMIL classification (Fig. 2C, D) and Graziani et al. regression (Fig. 2E, F) approaches exhibit a lack of clear distinction between predicted scores for HRD+ and HRD- patients. Notably, the CAMIL regression model displays a more pronounced separation

between the score distributions of HRD+ and HRD- patients in both the internal and external validation cohorts (Fig. 2G, H), compared to the other approaches. We further quantified this in all tumor entities and found that in all 7 of the selected TCGA cohorts, this distance was larger in CAMIL regression than in CAMIL classification, resulting in a greater class separability. In CPTAC, as compared to the CAMIL classification approach, class separability was improved in 2 out of 4 cohorts when using the CAMIL regression approach. Overall, our CAMIL regression approach improves the mean absolute separation distance of the groups' medians by 9.9% for the test set of the TCGA training cohort, and 4.9% for the external CPTAC test cohort compared to CAMIL classification (Supplementary Table 3). Compared to the

**Fig. 2 | Performance overview of classification versus regression approaches predicting the homologous recombination deficiency (HRD) score.**
**A, B** Boxplots representing the Area Under the Receiver Operating Characteristic (AUROC) values for HRD predictions. Predictions are made via three methods: I) CAMIL classification, II) Graziani et al. regression, and III) CAMIL regression. Models were tested using the internal datasets from The Cancer Genome Atlas (TCGA) and the external datasets from the Clinical Proteomic Tumor Analysis Consortium (CPTAC) effort. Cancer types included in these analysis are glioblastoma (GBM), pancreas adenocarcinoma (PAAD), endometrial cancer (UCEC), colorectal cancer (CRC), breast cancer (BRCA), lung adenocarcinoma (LUAD), and lung squamous cell cancer (LUSC). Non-significant AUROC values are represented as transparent violin plots. A two-sided DeLong's test was applied across all three architectures, with Bonferroni correction for multiple hypothesis testing (α = 0.0167). Source data, including the exact p-values, are provided as a Source Data file. **C–H** Depiction of the proportional distribution of normalized prediction scores. Normalization is performed to ensure a consistent scale for comparison across the different methods' prediction scores. The predicted scores are min-max normalized with 95% of the data falling in between the 2.5th and 97.5th percentile, removing extreme values that potentially distort the scaling. Plotted scores are from the internal test set of TCGA-UCEC and the external test set CPTAC-UCEC. The compared models are CAMIL classification, Graziani et al. regression, and CAMIL regression. Ground-truth classes are illustrated as a darker shade (HRD+) and a lighter shade (HRD−) of the color designated for the three tested model architectures, respectively. The sample size to plot the distributions is $n = 282$ and $n = 99$ independent patient samples for TCGA-UCEC and CPTAC-UCEC, respectively. The box plot represents the interquartile range (IQR), with the lower, middle and upper edge being the 25th, 50th, and 75th percentile. The whiskers of the box plots are defined as the minimum and maximum values 1.5 times the IQR away from the lower and upper quartiles of the data, respectively. Source data for the distributions and boxplots are provided as a Source Data file.

regression approach from Graziani et al. CAMIL regression improves the mean absolute separation distance of the groups' medians by 6.6% for the test set of the TCGA training cohort, and 9.5% for the external CPTAC test cohort (Supplementary Table 3).

Next, we compared CAMIL regression to Graziani et al.[32] regression by assessing the Pearson correlation coefficient (Pearson's r) of the predicted scores compared to the clinically-derived ground-truth scores. In TCGA, the CAMIL regression model reached higher Pearson's r scores than the Graziani et al.[32] model in all of the 7 selected cohorts (Supplementary Table 4). In the CPTAC validation cohort, the CAMIL regression model reached higher Pearson's r scores than the Graziani et al.[32] model in 2 out of 4 of the selected cohorts, LUSC and UCEC, while performing similarly poorly in PAAD (Supplementary Table 4). To determine the reason for our superior performance over Graziani et al.[32] regression (Supplementary Fig. 1), we conducted an ablation study of the CAMIL regression approach (Supplementary Table 5). These results revealed that the inferior performance in Graziani et al.[32] approach for predicting clinical biomarkers is mainly due to the standard stochastic gradient descent optimizer, compared to the stochastic gradient descent with adaptive moments optimizer in our CAMIL regression approach (Supplementary Table 6). Taken together, these data indicate that the CAMIL regression method outperforms the CAMIL classification and the Graziani et al.[32] regression approaches in learning more nuanced morphological patterns, as shown by the increased distance between prediction groups and consistently higher correlation coefficients, respectively.

Lastly, we proceeded to investigate the impact of somatic and germline mutations in *BRCA1/2* on HRD predictions derived from TCGA-BRCA. We detected a statistically significant disparity in HRD prediction groups for cases with *BRCA1* germline mutations ($p \leq 0.0001$) and *BRCA2* somatic mutations ($p \leq 0.05$) in all three modeling approaches. Conversely, no such significance was observed in HRD prediction groups for cases with *BRCA1* somatic and *BRCA2* germline mutations (Supplementary Fig. 2A–C). Additionally, we examined the concordance between HRD predictions from TCGA-CRC and the status of microsatellite instability (MSI) and tumor mutational burden (TMB). In the CAMIL regression approach, we observed a statistically significant difference in HRD prediction groups in relation to both MSI status ($p \leq 0.01$) and TMB status ($p \leq 0.05$). Therefore, a higher HRD prediction score from CAMIL regression is associated with microsatellite stable (MSS) and low TMB tumors within TCGA-CRC samples. Interestingly, such an association was not evident when using either the CAMIL classification or the Graziani et al. regression approaches (Supplementary Fig. 2D–F).

### Regression predicts key biological process biomarkers from histology

Having shown that our CAMIL regression method can predict HRD from histology WSIs, we expanded our experiments to additional biomarkers. We investigated biomarkers related to the three key components of solid tumors: tumor cells, stroma, and immune cells. For tumor cells, we aimed to predict proliferation, as measured by an RNA expression signature[36]. For stroma, we aimed to predict stromal fraction (SF), as assessed via DNA methylation analysis[36]. For immune cells, we investigated the tumor infiltrating lymphocytes regional fraction (TIL RF), the leukocyte fraction (LF), and the lymphocyte infiltration signature score (LISS)[36]. We found that our CAMIL regression method was able to predict all of these five biomarkers with high AUROCs across cancer types in the TCGA cohort (Supplementary Table 7). For example, in breast cancer, the AUROCs for these five biomarkers were 0.88 [0.86−0.91 in TIL RF, 0.84 [0.81−0.86] in proliferation, 0.80 [0.77−0.83] in leukocyte fraction, 0.80 in LISS, and 0.81 [0.78−0.83] in stromal fraction. In colorectal cancer, these AUROCs were 0.79 [0.75−0.84], 0.59 [0.51−0.66], 0.76 [0.72−0.81], 0.70 [0.66−0.74], 0.68 [0.63−0.73], respectively. Across all cancer types, AUROCs of above 0.70 were reached in 28 out of 34 models (Supplementary Table 7). These findings show that the regression-based DL model can be trained to predict tumor cell proliferation, stromal fraction and immune-cell-related biomarkers from H&E histopathology.

To further assess these results, we compared them to the state-of-the-art CAMIL classification approach using the AUROC as the evaluation metric, with 95% CI. Using site-aware splits, our proposed CAMIL regression reached higher AUROCs than the CAMIL classification in 29 out of 34 instances. Statistically, CAMIL regression outperformed CAMIL classification in 4 out of 34 instances, while the remaining cases showed no statistically significant difference in performance between the CAMIL classification and CAMIL regression approaches (Fig. 3B, Supplementary Tables 8 and 9). CAMIL regression outperformed CAMIL classification in TCGA-BRCA in two targets, LISS (0.80 [0.78−0.83], $p \leq 0.0167$) and TIL RF (0.88 [0.86−0.91], $p \leq 0.0167$). Moreover, CAMIL regression outperformed CAMIL classification in proliferation for TCGA-CRC (0.59 [0.51−0.66], $p \leq 0.01$) and TCGA-LIHC (0.87 [0.82−0.91], $p \leq 0.0167$).

Next, we measured the performance of CAMIL regression versus Graziani et al. regression. Our proposed CAMIL regression reached higher AUROCs than Graziani et al. regression in 33 out of 34 instances (Supplementary Tables 7 and 8), and higher Pearson's r in all 34 instances (Supplementary Figs. 3 and 4). CAMIL regression outperformed Graziani et al. regression in 14 out of 34 instances, whereas CAMIL classification outperformed Graziani et al. regression in only 5 out of 34 instances in a statistically significant manner (Supplementary Table 9). These findings collectively demonstrate that utilizing the CAMIL regression approach leads to an average 4% increase in the AUROCs compared to employing the CAMIL classification approach, and an average 12% increase as compared to employing the Graziani et al. regression approach for the same task of predicting key biological process biomarkers from histology (Supplementary Table 8).

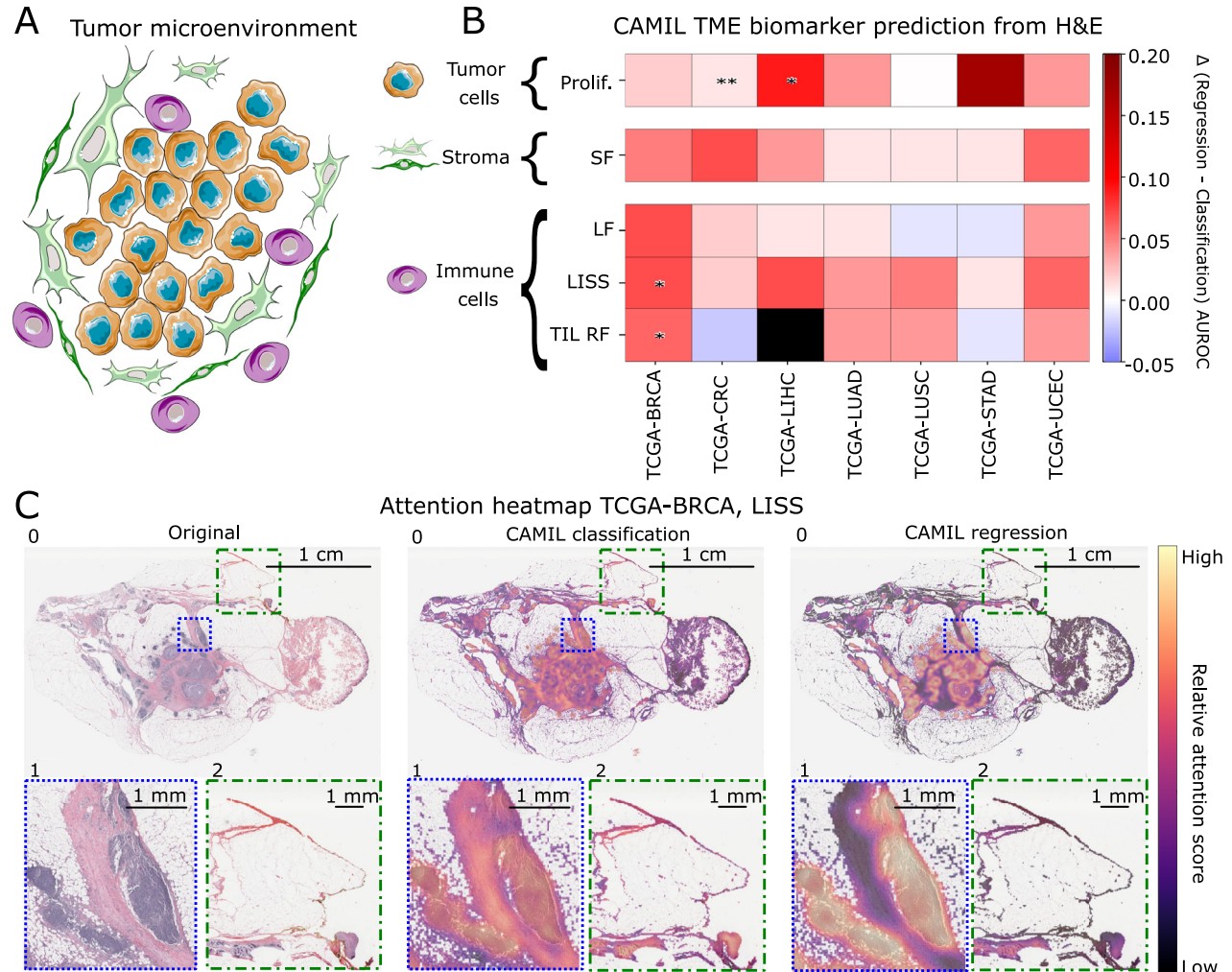

**Fig. 3 | CAMIL classification versus CAMIL regression for the prediction of continuous biological process biomarkers of the tumor microenvironment.** **A** Simplified depiction of the tumor microenvironment (TME) as the primary focus of our analysis, which includes tumor cells, stroma, and immune cells. **B** Heatmap indicates the deltas of Area Under the Receiver Operating Curve (AUROC) between CAMIL regression and CAMIL classification for five biological process biomarkers: tumor infiltrating lymphocytes regional fraction (TIL RF), proliferation (Prolif.), leukocyte fraction (LF), lymphocytes infiltrating signature score (LISS), and stromal fraction (SF). These biomarkers were tested on the sets of various cancer types including breast cancer (BRCA), colorectal cancer (CRC), liver hepatocellular carcinoma (LIHC), lung adenocarcinoma (LUAD), lung squamous cell cancer (LUSC), pancreas adenocarcinoma (PAAD), stomach cancer (STAD), and endometrial cancer (UCEC), which were all sourced from The Cancer Genome Atlas (TCGA) program for site-aware split folds. Higher positive delta indicates a superior performance by the CAMIL regression model. Asterisks denote statistical significance resulting from a paired two-tailed DeLong's test (α = 0.0167). **C** Representative attention heatmap of a slide from the TCGA-BRCA test set. Image 0 displays the entire slide, highlighting a diagnostic area of interest in Image 1. Image 2 represents an area containing presumably non-essential diagnostic information. This sequence is repeated for the original slide, the attention heatmap using CAMIL classification, and the attention heatmap using CAMIL regression for the LISS biomarker. Areas with higher attention scores are more critical for the model's decision-making. Source data are provided as a Source Data file. Parts of the figure were drawn by using pictures from Servier Medical Art. Servier Medical Art by Servier is licensed under a Creative Commons Attribution 3.0 Unported License (https://creativecommons.org/licenses/by/3.0/).

## Regression enhances correspondence to clinical knowledge in biomarker predictions from histology

Next, we investigated the capabilities of the correspondence to regions of known clinical relevance of the CAMIL classification model compared to the CAMIL regression model. For this, we evaluated the biological plausibility of spatial prediction heatmaps obtained by deploying both the LISS regression and the LISS classification models trained on tumors in the site-aware split test set of the TCGA cohort. Even though the LISS is only available as a weak label with one score per WSI, a robust model should still be capable of highlighting regions associated with the LISS, and these regions should predominantly contain lymphocytes. Indeed, we saw that both the classification model and the regression model placed their attention on lymphocyte-rich regions (Fig. 3C-0). Nevertheless, in the evaluated WSIs, the LISS regression model yielded a sharper delineation of lymphocyte-rich regions and placed less attention on areas where histologic features are less relevant. Contrastingly, the LISS classification model demonstrated relatively less confidence in areas with a dense lymphocyte population compared to the regression model, as indicated by a lower attention score (Fig. 3C-1). The classification model assigns importance to regions without any presumed clinical relevance, as evidenced by the fact that the model highlighted the tissue edge which lacks high density lymphocytes regions (Fig. 3C-2). Similar observations were made for the heatmaps produced by the Graziani et al. regression model (Supplementary Fig. 5), which emphasizes areas without presumed clinical relevance while overlooking lymphocyte-rich regions. We quantified these findings by a blinded review of 42 attention heatmaps from the CAMIL classification and CAMIL regression models by KJH, a pathology resident. Based on the expert review, the CAMIL regression approach produced attention heatmaps with better

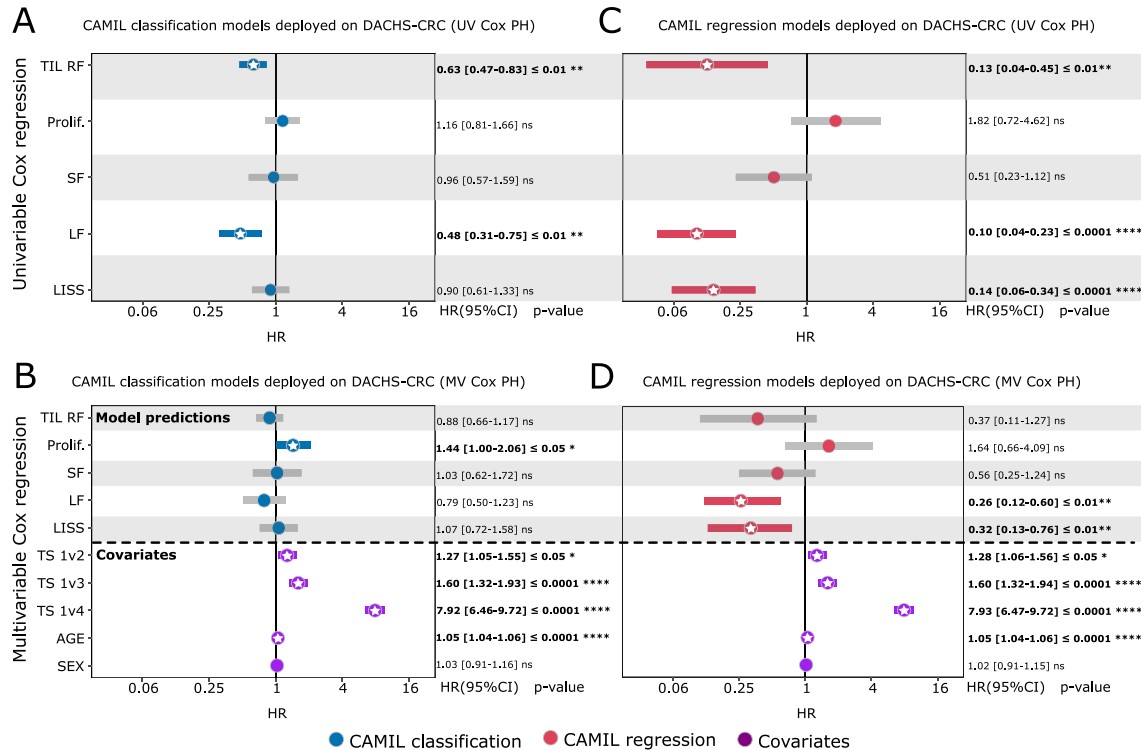

**Fig. 4 | Overview of the externally validated prognostic capabilities of the trained models to predict overall survival. A, B** Depiction of univariable (UV) and multivariable (MV) Cox proportional-hazards (PH) analyses of the CAMIL classification models. **C, D** Depiction of UV and MV Cox PH analyses of the CAMIL regression models. These models were trained on the biological process biomarkers from the breast cancer cohort from The Cancer Genome Atlas (TCGA) program and deployed on the external colorectal cancer (CRC) cohort from the Darmkrebs: Chancen der Verhütung durch Screening (DACHS) study. For the MV Cox PH analysis, each model's continuous output for the DACHS samples, from CAMIL classification and CAMIL regression, is independently considered alongside three covariates: tumor stage (TS), age, and sex. The observed biological process biomarkers include tumor infiltrating lymphocytes regional fraction (TIL RF), proliferation (Prolif.), leukocyte fraction (LF), lymphocyte infiltration signature score (LISS), and stromal fraction (SF). Stars indicate statistical significance ($p \leq 0.05$) for hazard ratios (HR) and their 95% confidence intervals (CI). The $p$-values and 95% CI are calculated through fitting the Cox's proportional hazard model for each variable independently. An HR confidence interval crossing 1 indicates non-significant prognostication capability. Prognostic capabilities that exhibit a stronger effect can be considered relatively better, as indicated by a HR further away from 1, printed in bold. The error bars are the 95% CI, with the measure of the centers being the estimated HR for each variable. The sample size to derive statistics is $n = 2297$ independent patient samples for each variable, with $n = 1345$ males (median age 69), $n = 952$ females (median age 70). Source data, including the exact p-values and disaggregated results by sex for the univariate Cox PH analysis, are provided as a Source Data file.

correspondence to regions of known clinical relevance in 34 out of 42 cases. In 6 out of 42 cases, the CAMIL classification approach was more favorable. Similar performance between the CAMIL classification and regression approaches was observed in 2 out of 42 cases. Hence, CAMIL regression outperforms CAMIL classification in 81% of cases based on blinded review. Taken together, these data demonstrate that the regression approach gives a statistically significantly better AUROC for the investigated biomarkers ($p \leq 0.0167$; Supplementary Tables 8 and 9), and a markedly improved capability of correspondence to regions of known clinical relevance, compared to the CAMIL classification and Graziani et al. regression approach (Supplementary Fig. 5).

**Regression-based biomarkers improve survival prediction in colorectal cancer**

Biological processes of tumor cell proliferation, deposition of stromal components, and infiltration by lymphocytes are biologically relevant during tumorigenesis and progression, and are known to be related to clinical outcome[37,38]. Thus, prediction of lymphocytic infiltration from H&E pathology slides should be relevant for prognostication. We investigated this in a large cohort of 2297 patients with colorectal cancer from the Darmkrebs: Chancen der Verhütung durch Screening (DACHS) study, for which H&E WSIs and long-term (10 years) follow-up data were available for overall survival (Supplementary Table 10).

First, we deployed the models that were trained on breast cancer patients in TCGA, as it is the only cancer type where CAMIL classification and CAMIL regression exhibited significantly different AUROCs ($p \leq 0.0167$) in 2 out of 5 biomarkers (Fig. 3B). We then deployed the CAMIL classifications models on WSIs from patients enrolled in DACHS. However, instead of utilizing the predicted label from the classification model, we employed the continuous scores for classification, i.e. logits [0,1], which allows for a more comparable survival analysis with the continuous regression scores. Upon following this approach, we assessed the prognostic impact of the predicted classification scores using univariable and multivariable Cox proportional-hazards models for overall survival (Fig. 4A, B), yielding hazard ratios (HR). In our analysis, significant risk-group stratification was observed in 2 out of 5 biomarkers by the classification models (Fig. 4A, Supplementary Table 11), namely TIL RF ($p \leq 0.01$) and leukocyte fraction ($p \leq 0.01$) exhibiting statistically significant findings. In the multivariable survival model (Fig. 4B, Supplementary Table 12), the classification models show significant prognostic capabilities in only 1 out of 5 biomarkers: proliferation ($p \leq 0.05$). This hazard ratio represents only a modest predictability of survival, as the CI of proliferation in the multivariable survival model touches the point of insignificance on its lower boundary with an HR of 1.44 [1.00–2.06]. After repeating the procedure with continuous scores obtained from the CAMIL regression models, we found that the regression models markedly improved

the survival prediction. The regression model demonstrated a significant risk-group stratification in 3 out of 5 biomarkers (Fig. 4A, Supplementary Table 11): TIL regional fraction ($p \leq 0.01$), leukocyte fraction ($p \leq 0.0001$) and LISS ($p \leq 0.0001$). Furthermore, in the multivariable survival model (Fig. 4B, Supplementary Table 12), the regression models exhibited significant prognostic capabilities in the same 2 biomarkers: leukocyte fraction ($p \leq 0.01$) and LISS ($p \leq 0.01$).

Second, we replicated the aforementioned experiments by deploying the TCGA colorectal cancer biomarker models on DACHS. These models demonstrated comparable performance metrics overall, with the exception of one biomarker (Fig. 3B). The CAMIL regression and classification models showed significant risk-group stratification in the majority of the biomarkers (Supplementary Table 11). When measuring the significant prognostic capabilities of the models through the HR and corresponding $p$-value, CAMIL regression yielded HR with a stronger effect compared to CAMIL classification in 3 out of 5 biomarkers (Supplementary Table 12): TIL regional fraction (0.33, $p \leq 0.05$), leukocyte fraction (0.24, $p \leq 0.05$) and LISS (0.07, $p \leq 0.0001$).

These observations provide evidence that CAMIL regression has effectively learned robust and generalizable prognostic features across diverse cohorts and cancer types, a capability that was not matched by CAMIL classification. Similarly, we conducted the same experiments for the Graziani et al. regression models (Supplementary Table 13). However, the Cox models failed to converge due to low variability observed in the patients' predictions. This outcome strongly suggests that the prognostic features learned by the regression model proposed by Graziani et al. lack the necessary generalizability when extended to external cohorts. Moreover, we disaggregated the univariable Cox proportional-hazard models for CAMIL classification and regression by sex (Supplementary Tables 14 and 15). Taken together, these data demonstrate that training models on biologically relevant biomarkers using weakly supervised learning result in CAMIL regression models that outperform the state-of-the-art classification and regression approaches in prognostication. This highlights the potential of CAMIL regression to enhance weakly supervised learning for clinical applications of DL systems.

## Discussion

Since 2018, the field of digital pathology has rapidly expanded to include the development of tools for predicting molecular biomarkers from routine tumor pathology sections, which has led to the development of clinically approved products. Traditional DL methods have limited the analysis of many biomarkers, including HRD and gene expression signatures, which are continuous values, by categorizing them into discrete classes. Our study provides direct evidence that regression networks, such as the CAMIL regression method described in this study, which builds on recent work using attention-based multiple instance learning and self-supervised pretraining of the feature extractor[18,20,33], outperforms traditional classification and regression networks in predicting these biomarkers. This approach unlocks a key clinical application area for pathology-based biomarker prediction.

Our proposed CAMIL regression approach has shown promising results in improving the accuracy and separability of biomarker predictions compared to CAMIL classification and Graziani et al.[32] regression. This improvement is particularly noticeable for biomarkers that have a clinically established threshold for categorization, such as HRD. Our observations of HRD predictions in relation to *BRCA1* and *BRCA2* somatic and germline mutations in TCGA-BRCA were consistent in all three modeling approaches, which align with a previous study on HRD[39]. Collectively, these findings affirm the ongoing need for formal germline testing in breast cancer. In our analysis of the predicted HRD scores using CAMIL regression in relation to MSI status and TMB status in TCGA-CRC, we discovered that CAMIL regression successfully identified correlations that align with established medical concepts[40,41]. These correlations remained undetected by the Graziani et al. regression and CAMIL classification approaches. Additionally, the paucity of HRD+ cases in TCGA-CRC ($n = 16$) suggests that CAMIL regression has the capacity to identify morphological HRD phenotypes with fewer patient samples compared to CAMIL classification and Graziani et al. regression. Similar improvements can be observed for biomarkers that do not have any clinically relevant cut-off point and would traditionally necessitate dichotomization for analysis, such as immune biomarkers. Moreover, our CAMIL regression approach demonstrates better generalization capabilities compared to both the regression approach by Graziani et al.[32] and CAMIL classification, as seen in the external test cohorts across multiple experiments. Of note, we identified that the optimizer used in Graziani et al.[32] predominantly caused the regression model to converge towards the mean, thereby explaining the observed discrepancy.

In addition, our study highlights the advantages of CAMIL regression-based biomarker prediction over CAMIL classification-based and Graziani et al. regression-based predictions in terms of the correspondence to regions of known clinical relevance. We demonstrated that, for tumor infiltrating lymphocytes, attention heatmaps generated through CAMIL regression were preferred in 81% of cases compared to those generated through CAMIL classification. CAMIL regression also resulted in an improvement in survival prediction based on immunologic biomarkers, as it allowed for more effective stratification of risk groups for overall survival compared to CAMIL classification models. The biomarkers were deliberately chosen on the basis of their prognostic capabilities[42–45], and are better reflected by the tumor morphology analysis through the CAMIL regression approach as compared to the CAMIL classification approach. Our CAMIL regression approach has consistently demonstrated superior prognostic capabilities, even when compared to the state-of-the-art CAMIL classification model. In contrast, when applying the Graziani et al. regression approach to the external cohort, it yielded predictions with exceedingly low variance, obstructing the Cox proportional-hazards model from converging, thereby further highlighting the limitations inherent in the Graziani et al. regression approach.

Our study brings valuable insights and contributions to the field, but it is not without its limitations. For example, the range of our experiments were limited to a select number of tumors and clinical targets, where not every analyzed clinical target had an external test set with the same clinical information available. This resulted in pseudo-external test sets through site-aware splits, and blind deployments on an external cohort. Additionally, none of the hyperparameters of the trained models were optimized. Further research could expand the analysis to a wider variety of cancers and clinical targets, while also exploring potential pitfalls of regression in computational pathology. Moreover, the analysis of continuous biomarkers, such as gene expressions, encounter various sources of noise and uncertainty in the measurements to define the ground-truth[46]. In such cases, relying solely on the exact values of the variable for prediction purposes can be problematic for training a regression model. Future work should consider the Kullback–Leibler divergence as a loss function[47] to deal with label noise of continuous biomarkers. By utilizing cut-offs, the prediction task is transformed into a seemingly simpler classification problem, at the cost of information loss[24]. The trade-off between noisy labels in regression versus loss of information in classification through dichotomization requires further research for the explicit delineation in which biological prediction task regression fails, and why. The approaches described here, however, provide a proof-of-principle for the use of regression-based attMIL systems and their potential impact for the inference of biomarkers and prediction of outcomes from histologic WSIs, expanding the repertoire of applications of DL in precision medicine.

## Methods

### Ethics statement

We examined anonymized patient samples from several academic institutions in this investigation. The collection and analysis of samples in the DACHS cohort was approved by the local ethics board. Written informed consent was obtained by participants in DACHS. Participants received no compensation for participation. CPTAC and TCGA did not require formal ethics approval for a retrospective study of anonymised samples. The overall analysis was approved by the Ethics commission of the Medical Faculty of the Technical University Dresden (BO-EK-444102022).

### Image data and cohorts

A total of 11,671 raw WSIs were scanned by an Aperio ScanSlide scanner and pre-processed in this study. Two types of clinical targets were analyzed to observe the performance of the classification and regression models: 1) continuous variables with a known clinically relevant cut-off for categorization, and 2) continuous variables with unknown clinically relevant cut-offs, thus requiring categorization by splitting at the median. These categories of targets were chosen due to theory mentioning the loss of information by splitting at the median[24], but does not mention the loss of information when utilizing clinically relevant cut-offs before training the model.

The target with a clinically relevant cut-off is homologous recombination deficiency (HRD) (Supplementary Table 16), a clinically relevant biomarker in solid tumor types, such as breast cancer. One way to calculate HRD is by adding up the three subscores, Loss of Heterozygosity (LOH), Telomeric Allelic Imbalance (TAI) and large-scale state transitions (LST), giving us a continuous value ranging from 0 to 103 in the training sets. A clinically relevant cut-off point of HRD ≥ 42 was used to binarize the continuous score[48].

The targets without a known clinically relevant cut-off point are biological process biomarkers (Supplementary Table 17), which are interesting to analyze due to their prominent role in immunotherapy outcome prediction[36,49,50]: Stromal Fraction (SF) with range [0, 0.92] and leukocyte fraction (LF) with range [0, 0.96] as assessed via DNA methylation analysis, lymphocyte infiltrating signature score (LISS) with range [−3.49, 4.17] and proliferation (Prolif.) with range [−2.86, 1.59], as measured by RNA expression data and tumor infiltrating lymphocytes regional fraction (TIL RF) with range [0, 63.65], quantified using a DL based classification. For TCGA-LIHC, there was no data available for TIL regional fraction, leading to an analysis of 5 targets in 7 cancer types with 5-fold cross-validation, resulting in (35-1)*5 models for each modeling type, of which the AUROC and 95% CI of the 5 folds per target and tumor type was reported.

### Model description

The entire image processing pipeline, from whole-slide image (WSI) to patient-level predictions, consisted of three main steps: 1) image pre-processing, 2) feature extraction, 3a) classification-based attention attMIL and 3b) regression-based attMIL for score aggregation resulting in patient-level predictions (Fig. 1A, B).

All WSI in the experiments were tessellated into image patches at a resolution of 224 by 224 pixels with an edge length of 256 μm, resulting in a Microns Per Pixel (MPP) value of approximately 1.14. After tessellation, every image patch underwent a rejection filter using the Canny edge detection method[51], removing blurry patches and the white background of the image when two or less edges were detected in the patches. The remaining patches were color-normalized in order to reduce the H&E-staining variance across patient cohorts according to the Macenko spectral matching technique[52], with a prior added step of brightness standardization. For pre-processing, our end-to-end WSI pre-processing pipeline was utilized. The target image used to define the color distribution was uploaded to the GitHub repository.

Every pre-processed image patch was turned into a 2048 feature vector through inference of a ImageNet-weighted ResNet50-based self-supervised contrastive clustering model fine-tuned on 32,000 WSIs from different cancer types; RetCCL[33]. The feature extraction resulted in an $(n \times 2048)$ feature matrix per patient, where n is the number of $(224 \times 224$ pixels) pre-processed image patches.

### Experimental setup and implementation details

For the experiments, 5-fold cross-validation on patient-level with site-aware splits was performed to train the models. Site-aware splits ensure that patients are stratified and grouped by the hospital the WSI originated from, creating a stratified random 80-20 split which forces all patients from the same hospital to exist in either the training and internal validation set, or the internal test set, while retaining ground-truth class distributions. Specifically, in TCGA, site-specific histological features were shown to be present in the WSI, causing biased evaluations in the model when not accounted for accordingly during the training procedure[34]. The basis for the weakly supervised classification and regression was adapted from the attention-based multiple instance learning (attMIL) method by Ilse et al.[53]. Our proposed model used Balanced MSE[54] as a loss function to account for the natural class imbalance in clinical settings, as well as the Adam optimizer[55] and an attention component followed by a MLP head[53] which was trained for 25 epochs. The dropout layer was removed, due to loss of performance in regression in tabular data settings[56]. The attMIL variant in our proposed CAMIL regression differs from Ilse et al. by swapping their feature extractor with a pre-trained ResNet50 with ImageNet weights, fine-tuned on 32,000 histopathology images in a self-supervised manner using contrastive clustering shown to yield significantly better results on WSI image analysis[33]. Moreover, the classification head consisting of a fully-connected (FC) layer and sigmoid operation was swapped with custom heads to allow for classification and regression tasks to be performed. The attention component was not altered.

To evaluate the relative supremacy between classification and regression, first, the CAMIL regression method was compared with 1) the regression method from Graziani et al. and 2) the CAMIL classification method on the continuous HRD score and clinically-relevant binarized HRD score, respectively. Similarly, CAMIL regression was compared to CAMIL classification and Graziani et al. regression on continuous biomarkers related to biological processes with no known clinically-relevant cut-off points, where the median score per target was used for binarizing. Moreover, an expert review by a pathology resident was conducted on attention heatmaps produced by CAMIL classification and CAMIL regression to determine which method yielded the heatmaps with the best correspondence to regions of known clinical relevance. Finally, the prognostic capabilities of CAMIL regression, CAMIL classification and Graziani et al. regression was evaluated on an external data cohort DACHS-CRC by predicting survival of groups stratified by the models which were trained on the same biological process biomarkers and extracted features. For the HRD scores, the models were trained on TCGA-BRCA, TCGA-CRC, TCGA-GBM, TCGA-LUAD, TCGA-LUSC, TCGA-PAAD, TCGA-UCEC and externally validated on CPTAC-LUAD, CPTAC-LSCC, CPTAC-PDA and CPTAC-UCEC. For the biological process biomarkers, the models were trained on TCGA-BRCA, TCGA-CRC, TCGA-LUAD, TCGA-LUSC, TCGA-LIHC, TCGA-STAD and TCGA-UCEC. Every biomarker prediction model that was compared, both regression and classification, consisted of the exact same patients for training, internal validation, internal testing and external testing (Supplementary Tables 16 and 17).

For the regression method from Graziani et al. we introduced the self-supervised component as feature extractor[33] followed by embedding-level attention aggregation, instead of the ImageNet weighted ResNet18 backbone followed by patch-level attention aggregation in the original study by Graziani et al.[32]. As it was shown that the self-supervised backbone increases performance and

generalizability compared to an ImageNet weighted architecture as backbone[33], we added the self-supervised component in order to compare the regression heads in isolation. The commonalities between the models are the learning rate (1.00E-04), weight decay (1.00E−02), patience (12 epochs), the attention component[53] and the fit-one-cycle learning rate scheduling policy[57]. The differences of the models' hyperparameters and optimization strategies (Supplementary Table 5) of Graziani et al. and our CAMIL regression model were broken down in an ablation study to find the reason for the performance differences of the regression heads.

To explicitly denote the terminology in this paper, we define training as performing 5-fold cross-validation, thus 5 training iterations of $n$ epochs, on the respective cohorts of TCGA with corresponding molecular biomarkers. As previously described, training is executed with 5 times an 80-20 split of the TCGA patients, resulting in 5 models trained and tested on different subsets of the training cohort. This 20% test set allows for the evaluation of the selected architectures' models on the TCGA cohort. We define retraining as performing a single training iteration of $n$ epochs on 100% of the TCGA cohort. Consequently, no test set remains to evaluate the performance on the training cohort TCGA. We define internal validation of the models as applying the models on a subset of data that was unseen during training, but still belongs to the same cohort of patients from TCGA. We define external validation as deploying the models on a subset of data that was unseen during training, given that the subset of data comes from a different submission site than the training cohort, such as CPTAC and DACHS. External validation is important in order to test the generalizability of the trained models to different environments in which the pathology slide and corresponding biomarker information is prepared. For the statistics, we use an ensemble of the predictions resulting from the 5-fold models being applied to the test sets. Thus, we obtain multiple scores for each patient, which are then aggregated using the median. This results in 1 ensembled score for each patient, now considered an independent sample, which is used in subsequent statistical analyses. The effectiveness of the models' decision-making, as indicated by their attention heatmaps, is evaluated through the models' ability to correspond predictions to clinically relevant regions. Note that this evaluation concept should not be conflated with the comparison of interpretability capabilities across models[58], as the tested modeling approaches employ an identical attention mechanism to facilitate interpretability.

## Statistics and endpoints
The classification and regression method were made comparable in a similar dimension by utilizing the area under the receiver operating characteristic (AUROC) metric. For the definition of the binarized groups required for the AUROCs, the clinically-relevant cut-off for HRD was used, while for the biological process biomarkers, the continuous targets were split at the median. The prediction scores of the classification model [0,1] and the predictions of the regression models (−∞,∞) were used as continuous score for all the possible thresholds of the AUROC[59]. By utilizing this approach, it was possible to test which type of model, when provided with the same ground-truth binarized label, had the least overlap between the predicted score distributions for different groups. This, in turn, resulted in achieving the highest AUROC. However, the AUROC measures only the separation of groups' score distributions, but does not account for the distance between the distributions. Therefore, to determine whether there is an increased distance between distributions, the median and interquartile range (IQR) were calculated for the clinically relevant HRD+ and HRD- groups. However, this calculation was not performed for the biological process biomarkers due to the unclear relevance of distance between the dichotomized groups.

To determine statistical significance of the AUROCs, the 95% confidence interval (CI) of the 5 training folds was calculated for each model. For the statistical analyses, a median ensemble of predictions across the 5 folds was preferred instead of retraining a single model on the entire training cohort. This approach offers a more consistent generalizability to external cohorts (Supplementary Fig. 6), as also observed in other studies in computational pathology[15,60]. In order to identify if the AUROCs of the three compared models (CAMIL classification, regression from Graziani et al. and our proposed CAMIL regression) had a significant difference for the HRD target, three paired two-tailed DeLong's tests were performed for each cancer type. Similarly, the AUROCs of the biological process biomarkers' models were also compared using three paired two-tailed DeLong's tests performed for each cancer type. To account for multiple hypothesis testing, the $p$-values were adjusted through a Bonferroni correction ($\alpha = 0.0167$). For comparisons between the regression approaches, the Pearson's r and corresponding $p$-values were calculated using the median ensemble of predictions, resulting in a single aggregated prediction score for each patient.

To determine the prognostic capabilities of the biological process biomarkers' models, survival prediction analysis was done on an external cohort, DACHS. All 5 models trained through site-aware splits were blindly deployed, of which the median of the predicted scores were used for further analysis. The univariable and multivariable Cox proportional-hazards analysis were independently performed to determine the Hazard Ratio (HR) of the classification and regression models' predictive biomarker. The continuous score from the regression and classification models were used for the Cox proportional-hazards analyses, enabling a comparison between the survival models. The prognostic capabilities of the classification and regression models were independently analyzed together with three covariates: age (continuous, $\mathbb{R}^+$), sex (male versus female) and tumor stage (stage 1 versus 2, stage 1 versus 3, stage 1 versus 4). Thus, one model's continuous scores per target and the three covariates were analyzed for each model independently. Statistical significance of the HR is reached when the 95% CI does not cross a HR = 1, with models yielding HR further away from 1 indicating a stronger effect and thus defined as having better prognostic capability.

## Visualization and explainability
To compare the separability of the models' score distribution for HRD at a similar scale, all values for all three models were normalized individually to redistribute every model's score output between [0,1]. The method used for rescaling the predictions is a variation of min-max normalization with robust scaling, where min-max normalization is performed on 95% of the data falling in between the 2.5th and 97.5th percentiles of the predictions for each method. This procedure removes the extreme values in the predictions for each of the classification and regression methods, enabling us to calculate the separation for 95% of the remaining data. Consequently, we reduce the risk of extreme values affecting the scaling, and it focuses the performance comparison on the central portion of the data distribution. The robust min-max normalization method is only applied for the calculation and visualization of the separability, and does not affect any other reported metrics. To explain the classification and regression models' capability of decision-making using clinically relevant features, the attention component from the attMIL model architecture was utilized. The attention heatmaps were created by loading the attMIL model architectures for classification and regression into a fully convolutional equivalent[61] with their respective weights from the training procedure, which allows for a high-resolution attention heatmap, rather than 224×224 patches the model was trained on. By running inference on the WSIs of the patient, the attention layer which resulted from the patient-wise prediction was extracted and used as an overlay on the WSI to indicate hot zones which the model used in decision making. The TCGA-BRCA cohort was chosen for visualization to observe the contrast between similar- and superior performance of the regression model compared to the classification model in lymphocyte-based

targets. For each target, the classification and regression model were trained, validated and tested on the exact same patients using site-aware splits. The attention heatmaps for the blinded review were generated from patients with the top 42 highest expression of the LISS biomarker from the unseen internal TCGA-BRCA test set through the trained CAMIL classification and CAMIL regression models, resulting in 84 heatmaps in total. The models' capability of corresponding predictions to regions of known clinical relevance was reviewed by a pathologist, choosing the most accurate attention heatmap for each of the 42 patients. The attention heatmaps generated by the Graziani et al. regression models were excluded from the pathologist review due to initial observations that indicated unsatisfactory performance in both quantitative metrics (Supplementary Fig. 4) and the quality of the generated heatmaps (Supplementary Fig. 5).

### Reporting summary
Further information on research design is available in the Nature Portfolio Reporting Summary linked to this article.

## Data availability
The slides for TCGA are available at https://portal.gdc.cancer.gov/. The slides for CPTAC are available at https://proteomics.cancer.gov/data-portal. The molecular data for TCGA and CPTAC are available at https://www.cbioportal.org/ and additional biomarker data is available from Thorsson et al.[36]. The slides and biomarker data for DACHS were generated for prior studies[62–64] with restricted access. Biomarker data for DACHS are available by requesting Authorized Access to the phs001078 [https://www.ncbi.nlm.nih.gov/projects/gap/cgi-bin/study.cgi?study_id=phs001113.v1.p1] study. Applications for access to DACHS biomarker data are reserved for Senior Investigators and NIH Investigators as defined in https://dbgap.ncbi.nlm.nih.gov/aa/wga.cgi, and upon successful application grants access to the data for 1 year with the option to renew access. The slides for DACHS can only be requested directly through the DACHS principal investigators. The contact details are listed at http://dachs.dkfz.org/dachs/kontakt.html. The data generated in this study for the creation of the figures are provided in the Source Data file. Source data are provided with this paper.

## Code availability
All source codes are available under an open-source license on GitHub. The pre-processing pipeline is found at https://github.com/KatherLab/end2end-WSI-preprocessing/releases/tag/v1.0.0-preprocessing, the classification pipeline is found at https://github.com/KatherLab/marugoto/releases/tag/v1.0.0-classification, the regression pipeline is found at https://github.com/KatherLab/marugoto/releases/tag/v1.0.0-regression, and the classification and regression attention heatmaps are found at https://github.com/KatherLab/highres-WSI-heatmaps/releases/tag/v1.0.0-heatmaps.

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

## Acknowledgements

JNK is supported by the German Federal Ministry of Health (DEEP LIVER, ZMVI1-2520DAT111) and the Max-Eder-Programme of the German Cancer Aid (grant #70113864), the German Federal Ministry of Education and Research (PEARL, 01KD2104C; CAMINO, 01EO2101; SWAG, 01KD2215A; TRANSFORM LIVER, 031L0312A), the German Academic Exchange Service (SECAI, 57616814), the German Federal Joint Committee (Transplant.KI, 01VSF21048) the European Union (ODELIA, 101057091; GENIAL, 101096312) and the National Institute for Health and Care Research (NIHR, NIHR213331) Leeds Biomedical Research Centre. The views expressed are those of the author(s) and not necessarily those of the

NHS, the NIHR or the Department of Health and Social Care. J.S.R.-F. reports receiving personal/consultancy fees from Goldman Sachs, Bain Capital, REPARE Therapeutics, Saga Diagnostics and Paige.AI, membership of the scientific advisory boards of VolitionRx, REPARE Therapeutics and Paige.AI, membership of the Board of Directors of Grupo Oncoclinicas, and ad hoc membership of the scientific advisory boards of Astrazeneca, Merck, Daiichi Sankyo, Roche Tissue Diagnostics and Personalis, outside the scope of this study. UCLH Biomedical research centre is funded by the National Institute for Health Research (BRC399/NS/RB/101410). S.B. is also supported by the Department of Health's NIHR Biomedical Research Centre's funding scheme. T.M. was supported by The Brain Tumour Charity (GN-000389 clinical research training fellowship) and by the National Institute of health research (NIHR) with clinical lecturer fellowship (CL-2019-19-001). F.R.K. is supported by the Add-on Fellowship of the Joachim Herz Foundation. Q.Z. is funded by the China Scholarship Council (Grant n°201908070052). The DACHS study (H.B. and M.H.) was supported by the German Research Council (BR 1704/6-1, BR1704/6-3, BR1704/6-4, CH117/1-1, HO5117/2-1, HO5117/2-2, HE5998/2-1, HE5998/2-2, KL2354/3-1, KL2354/3-2, RO2270/8-1, RO2270/8-2, BR1704/17-1, and BR1704/17-2), the Interdisciplinary Research Program of the National Center for Tumor Diseases (NCT; Germany), and the German Federal Ministry of Education and Research (01KH0404, 01ER0814, 01ER0815, 01ER1505A, and 01ER1505B).

## Author contributions

OSMEN and JNK designed the study. OSMEN, MVT and MG developed the software. OSMEN, CMLL, TY, MH, HB, AB and JNK contributed to data collection and assembly. OSMEN, ZIC, CMLL, KJH, FRK, HSM, QZ, JC, NOB and J.S.R.-F interpreted and analyzed the data. All authors substantially contributed to writing and reviewing the report, approved the final version for submission, and have agreed to be personally accountable for the author's own contributions and to ensure that questions related to the accuracy or integrity of any part of the work, even ones in which the author was not personally involved, are appropriately investigated, resolved, and the resolution documented in the report.

## Funding

## Competing interests

O.S.M.E.N. holds shares in StratifAI GmbH. J.N.K. declares consulting services for Owkin, France; DoMore Diagnostics, Norway and Panakeia, UK; furthermore, J.N.K. holds shares in StratifAI GmbH and has received honoraria for lectures by Bayer, Eisai, MSD, BMS, Roche, Pfizer and Fresenius. J.S.R.-F. is funded in part by the Breast Cancer Research Foundation, by a Susan G Komen Leadership grant, and by the NIH/NCI P50 CA247749 01 grant. The mentioned competing interests are related to cancer and the computational analysis of histopathology slides, which is the main topic of this research. The remaining authors declare no competing interests.

## Additional information

¹Else Kroener Fresenius Center for Digital Health, Medical Faculty Carl Gustav Carus, TUD Dresden University of Technology, Dresden, Germany. ²Department of Medicine 1, University Hospital and Faculty of Medicine Carl Gustav Carus, TUD Dresden University of Technology, Dresden, Germany. ³Department of Visceral, Thoracic and Vascular Surgery, University Hospital and Faculty of Medicine Carl Gustav Carus, TUD Dresden University of Technology, Dresden, Germany. ⁴University of Applied Sciences of Western Switzerland (HES-SO Valais), Rue du Technopole 3, 3960 Sierre, Valais, Switzerland. ⁵Centre d'Histologie, d'Imagerie et de Cytométrie (CHIC), Centre de Recherche des Cordeliers, INSERM, Sorbonne Université, Université Paris Cité, Paris, France. ⁶Assistance Publique-Hôpitaux de Paris, Département de Pathologie, CHU Henri Mondor, F-94000 Créteil, France. ⁷Institute of Pathology, University Hospital RWTH Aachen, Aachen, Germany. ⁸Center for Integrated Oncology Aachen Bonn Cologne Duesseldorf (CIO ABCD), Cologne, Germany. ⁹Division of Clinical Epidemiology and Aging Research, German Cancer Research Center (DKFZ), Heidelberg, Germany. ¹⁰Division of Preventive Oncology, German Cancer Research Center (DKFZ) and National Center for Tumor Diseases (NCT), Heidelberg, Germany. ¹¹German Cancer Consortium (DKTK), German Cancer Research Center (DKFZ), Heidelberg, Germany. ¹²Institute of Pathology, University Hospital Heidelberg, 69120 Heidelberg, Germany. ¹³Tissue Bank, National Center for Tumor Diseases (NCT), University Hospital Heidelberg, 69120 Heidelberg, Germany. ¹⁴Department of Pathology and Laboratory Medicine, Memorial Sloan Kettering Cancer Center, New York, NY, USA. ¹⁵Pathology & Data Analytics, Leeds Institute of Medical Research at St James's, University of Leeds, Leeds, United Kingdom. ¹⁶Medical Oncology, National Center for Tumor Diseases (NCT), University Hospital Heidelberg, Heidelberg, Germany. ✉e-mail: jakob_nikolas.kather@tu-dresden.de

