## [Peer Review File · Nature Communications]

REVIEWER COMMENTS

Reviewer #1 (Remarks to the Author): Expert in cancer digital pathology and machine learning

El Nahhas et al. present a deep learning based biomarker predictor from histopathology whole slide images using a weakly supervised regression based approach with self-supervised attention network as the backbone. The method is applied for continuous biomarker prediction in nine cancer types and compared to categorical prediction approach. The results suggest that the proposed regression method slightly outperforms an existing regression method, as well as a classification based method. On general, the work has merit: state-of-the-art methods are used, and although the improvement to the compared methods is to some extent incremental, the extensive experimental work confirms that the difference is meaningful.

Specific comments:

-Intro: other examples exist in the literature towards prediction of continuous target variables from histopathology, including molecular level readouts - while true that majority of the present literature concentrate on classification approach of categorical variables, the authors could do better in covering prior work.

-The proposed CAMIL-regression method is compared to the regression based method by Graziani et al. Based on the presented results (Fig. 2), I would rather see the comparison between the two regression based methods or between all three (CAMIL reg, CAMIL classification, Graziani) being carried out in the subsequent experiments.

-The authors do discuss the challenge of using cut-offs for continuous variables, for certain readouts there are clinically relevant cut-offs but not for all, and that the cut-off leads to information loss. However, the other side of the coin is that the continuous variable is often very noisy, and prediction of specific values may be highly ambiguous depending on the variable. Consider expanding the discussion of the nature of the task.

-p5 r203: "did not reach a statistically significant AUROC in either classification or regression" -- difference, improvement?

Reviewer #2 (Remarks to the Author): Expert in colorectal cancer clinical research and genetics, and homologous repair deficiencies

The paper hypothesises that regression based deep learning will perform better than a classification one - a hypothesis that is very sound given that we do measure most biomarkers in a continuous manner. The authors should be congratulated in how they have written the paper and the detailed experimental method explanations - not always there in deep learning papers explained in a way that can be understood to a wider readership. The results of the regression based modelling performing better is not surprising and the authors have described these results well.

Strengths of the paper

- novel concept that is well supported by the evidence
- sample size of images used
- pan cancer
- external validation that supports the findings

Limitations -

given that the TCGA dataset is used - data on how the somatic/ germline results affected the results eg BRCA1/2 in breast and ovarian cancer; MMR in CRC and how their model performed in addition to identifying HRD important other markers like tumor mutation burden as this would give us an idea of perhaps the eventual utility of such a method for replacing the need for formal tumor/germline testing etc . It would certainly be on the minds of the reviewers and would be expected from a paper aiming for Nature communications . For the external validation cohorts - there would have been also treatment data available - how did the regression based DL identified e.g HRD cases correlate with treatment response ?

Reviewer #3 (Remarks to the Author): Expert in biostatistics and biomarkers

General comments:

I recognize that a huge amount of effort went into this manuscript; however, the statistical methods used were less than satisfactory.

1. AUC's confidence interval is asymmetrical unless you use a Wald's CI. Authors should justify why they choose the Wald's CI since it does produce a confidence interval exceed 0 and 1. For example,

supplemental Table 1, CTPAC-UCEC row under AUROC column, the upper confidence limit is 1.03 which is impossible for AUC.

2. The repeated ANOVA and pair t-test are not appropriate statistical methods in this context.

2.1. I understand that each cross-validation generate an AUC but these 5 AUC are not independent since the internal training dataset are not independent; therefore, repeated AUC is not an appropriate method to use. The 5 AUC can be averaged for a general AUC and the corresponding confidence interval can be produced by CVAUC R package.

2.2. Similar rationale applied to paired t-tests.

2.3. After you have the final prediction model, it can/should be applied to the whole training dataset to come up with the final model.

2.4. If you have one final model from the training dataset then it can be applied once to the external validation dataset which will produce one AUC; therefore, repeated ANOVA is not needed for the external validation dataset. To compare AUC across different model using external validation dataset, DeLong's pair test for AUC can be used.

3. I am puzzled by the extremely significant p-values reported in the appendix. For example, given that the training dataset has about 3000 patients and a R-square of 0.13, the p-value is reported to be 7.44E-18 which seems wrong. I wonder whether the authors lump the 5 cross-validation together and treated them as independent samples and calculate the R-square. If so, that is not proper since they are not independent samples.

Specific comments:

1. The stage variable in the Cox model must be modeled as a categorical variable since the risk of death of a patient with stage 4 cancer is definitely more than 4 times of the risk of a patients with stage 1 cancer.

2. Line 451, it should be "multivariable" rather than "multivariate". This comment applies to the whole paper.

3. Figure 4 can be confusing.

3.1. I assume that one model is fitted for categorical prediction with covariates and a separate model is fitted for regression prediction with covariates. If my understanding is correct then the covariates should have different coefficient but the covariates are reported only once on each sub-figure.

3.2. Given a feature, for example, Prolif in Figure 4d, it is difficult to know which variable is more important since they have similar p-value. Given one is categorical variable and the other is continuous, the coefficients are not comparable.

REVIEWER COMMENTS

Reviewer #1 (Remarks to the Author): Expert in cancer digital pathology and machine learning

Comment: El Nahhas et al. present a deep learning based biomarker predictor from histopathology whole slide images using a weakly supervised regression based approach with self-supervised attention network as the backbone. The method is applied for continuous biomarker prediction in nine cancer types and compared to the categorical prediction approach. The results suggest that the proposed regression method slightly outperforms an existing regression method, as well as a classification based method. On general, the work has merit: state-of-the-art methods are used, and although the improvement to the compared methods is to some extent incremental, the extensive experimental work confirms that the difference is meaningful.

Response: Thank you for your positive assessment of our work and the constructive feedback you have provided. We have now experimentally addressed each of the comments you put forward. Please see detailed responses below.

Comment: -Intro: other examples exist in the literature towards prediction of continuous target variables from histopathology, including molecular level readouts - while true that majority of the present literature concentrate on classification approach of categorical variables, the authors could do better in covering prior work.

Response: Thank you for bringing this to our attention. We have revised the introduction, and added a total of seven references to related studies. An example includes investigating regression approaches for gene expressions from breast cancer histology, as presented in the work of Mondol et al., 2023, *Cancers*. These additional sources not only provide further context for our experimental designs and analysis, but also help position our manuscript alongside the current state of the art.

Comment: -The proposed CAMIL-regression method is compared to the regression based method by Graziani et al. Based on the presented results (Fig. 2), I would rather see the comparison between the two regression based methods or between all three (CAMIL reg, CAMIL classification, Graziani) being carried out in the subsequent experiments.

Response: Thank you for providing these valuable suggestions, which upon addressing, have significantly enhanced the quality of this paper. We have now performed experiments for all three approaches: CAMIL regression, CAMIL classification, and Graziani et al. regression. The results consistently indicate that CAMIL regression outperforms Graziani et al. regression across all the newly

proposed experimental approaches. We have added these new findings in the result (p5, r144-156; p6, r203-211 and r227-229; p7, 282-286) and discussion (p8, r304, r325 and r334-337) section. Additionally, we have updated Figure 2, Suppl. Figure 3, 5, 6 and Suppl. Table 3, 7, 8, 9 and 13 to reflect these new changes. Moreover, we have adequately adjusted the p-values for our statistical tests across the three approaches, reducing the likelihood of random results, strengthening the performance efficacy for the CAMIL regression.

Comment: -The authors do discuss the challenge of using cut-offs for continuous variables, for certain readouts there are clinically relevant cut-offs but not for all, and that the cut-off leads to information loss. However, the other side of the coin is that the continuous variable is often very noisy, and prediction of specific values may be highly ambiguous depending on the variable. Consider expanding the discussion of the nature of the task.

Response: Thank you for raising this relevant and important point. In line with your suggestion, we have expanded the discussion section (p9, r345-353) to take into account noisy targets and the complexity associated with attempting specific predictions. Moreover, we have also incorporated a new reference that outlines potential future research directions, which could account for the prediction of the noisy continuous variables in regression and possibly aid in addressing these challenges.

Comment: -p5 r203: "did not reach a statistically significant AUROC in either classification or regression" -- difference, improvement?

Response: Thank you for pointing out this ambiguity. This has been resolved as a direct consequence of redoing our entire statistical analysis based on other reviewers' comments. We made our statistical tests (and the hypotheses being tested) more explicit in the methods (p12, 477-486).

Reviewer #2 (Remarks to the Author): Expert in colorectal cancer clinical research and genetics, and homologous repair deficiencies

Comment: The paper hypothesizes that regression based deep learning will perform better than a classification one - a hypothesis that is very sound given that we do measure most biomarkers in a continuous manner. The authors should be congratulated in how they have written the paper and the detailed experimental method explanations - not always there in deep learning papers explained in a way that can be understood to a wider readership. The results of the regression based modeling performing better is not surprising and the authors have described these results well.

Response: Thank you for your kind words and generous feedback regarding our work. It is our mission to put our best efforts in providing explanations for complex subjects in an understandable and engaging narrative. Knowing that our

manuscript has been able to convey this serves as great encouragement for our team.

Comment: Strengths of the paper

- novel concept that is well supported by the evidence
- sample size of images used
- pan cancer
- external validation that supports the findings

Response: Thank you for the positive assessment of our work! We have addressed all your comments below.

Comment: Limitations - given that the TCGA dataset is used - data on how the somatic/germline results affected the results eg BRCA1/2 in breast and ovarian cancer; MMR in CRC and how their model performed in addition to identifying HRD important other markers like tumor mutation burden as this would give us an idea of perhaps the eventual utility of such a method for replacing the need for formal tumor/germline testing etc. It would certainly be on the minds of the reviewers and would be expected from a paper aiming for Nature communications. For the external validation cohorts - there would have also been treatment data available - how did the regression based DL identified e.g HRD cases correlate with treatment response ?

Response: We are grateful for your suggestion and fully agree that replacing germline testing would be a highly relevant use-case. However, we only have limited germline data available for the TCGA-BRCA cohort, which we were able to obtain through our partners at Memorial Sloan Kettering. Consequently, we added the results of our analysis regarding the HRD predictions with *BRCA1/2* germline/somatic mutations in breast cancer, as well as MSI status and TMB status in colorectal cancer. The results are visualized in suppl. Figure 6, described in the results (p5, r164-175) and elaborated on in the discussion (p8, r306-315), which are in line with recently published works (Farmanbal et al., 2023, Nature Scientific reports; Budczies et al., 2022, Journal of Pathology: Clinical Research) and are added to the manuscript. Regarding the treatment response to PARP inhibitors and correlation with HRD, unfortunately this treatment data does not exist for these datasets. At the moment we are collecting clinical trial data to perform these types of studies. Our present study is an invaluable foundation for these follow-up studies by our team and other research groups, who can benefit from the open-source proposed methods for their own studies.

Reviewer #3 (Remarks to the Author): Expert in biostatistics and biomarkers

Comment: General comments:

I recognize that a huge amount of effort went into this manuscript; however, the statistical methods used were less than satisfactory.

Response: We highly appreciate your critical feedback on the methodological aspects of our presented work. We have now meticulously addressed and resolved each point you have brought to our attention. Please see our detailed responses below.

Comment: 1. AUC's confidence interval is asymmetrical unless you use a Wald's CI. Authors should justify why they choose the Wald's CI since it does produce a confidence interval exceed 0 and 1. For example, supplemental Table 1, CTPAC-UCEC row under AUROC column, the upper confidence limit is 1.03 which is impossible for AUC.

2. The repeated ANOVA and pair t-test are not appropriate statistical methods in this context.

2.1. I understand that each cross-validation generate an AUC but these 5 AUC are not independent since the internal training dataset are not independent; therefore, repeated AUC is not an appropriate method to use. The 5 AUC can be averaged for a general AUC and the corresponding confidence interval can be produced by CVAUC R package.

2.2. Similar rationale applied to paired t-tests.

Response: We deeply appreciate your expert input and detailed examination of our statistical approach. In response to your suggestions, we have thoroughly revised our statistical analysis. More details on these changes can be found in the responses below. In line with your suggestion, we have utilized the CVAUC R package to create the AUC and corresponding 95% confidence interval. These changes are now reflected throughout the entire paper: the results (p4, r101-103, r105-107, r130-133; p5, r185-189; p6, r199-201), suppl. Table 1,6 and 7, and suppl. Figure 2..

Comment: 2.3. After you have the final prediction model, it can/should be applied to the whole training dataset to come up with the final model.

2.4. If you have one final model from the training dataset then it can be applied once to the external validation dataset which will produce one AUC; therefore, repeated ANOVA is not needed for the external validation dataset. To compare AUC across different model using external validation dataset, DeLong's pair test for AUC can be used.

Response: Thank you for your insightful comment and the suggestion to employ DeLong's pair test for AUC. We have now incorporated your suggestions into the manuscript. In order to create a final prediction model, we first trained 5 instances on TCGA using cross-validation and subsequently deployed these instances on CPTAC across all the observed cancer types, biomarkers, and the three model architectures which are compared in our work. A simple ensemble model was then created by taking the median score of the 5 predictions of the external CPTAC cohort, leaving us with 1 prediction per patient, for which we then performed the suggested statistical test. Furthermore, we tested the retraining on the full cohort as suggested, the comparison of which can be found in Suppl. Figure 6. Based on these results, and examples from the literature using simple ensemble models (Campanella et al., 2019, Nature Medicine; Lu et al., 2021, Nature Biomedical Engineering), we opted for the median ensemble model to perform our

statistical analyses. As a result, the following additions and changes were made: Figure 1, 2, 3, 4, suppl. Table 2, 6, 7, 9, 11, 12, 13, Suppl. Figure 2, updated all the numbers throughout the results section, and updated the methods section.

Comment: 3. I am puzzled by the extremely significant p-values reported in the appendix. For example, given that the training dataset has about 3000 patients and a R-square of 0.13, the p-value is reported to be 7.44E-18 which seems wrong. I wonder whether the authors lump the 5 cross-validation together and treated them as independent samples and calculate the R-square. If so, that is not proper since they are not independent samples.

Response: Thank you for the detailed remark. We have fixed this through using the median ensemble method as mentioned above, resulting in 1 score per patient. Hence, we have changed the following in the manuscript: suppl. Table 4, 6, and suppl. Figure 2, 4, 5.

Comment:

1. The stage variable in the Cox model must be modeled as a categorical variable since the risk of death of a patient with stage 4 cancer is definitely more than 4 times of the risk of a patients with stage 1 cancer.

Response: Thank you for your comment. As per your suggestion, we have now implemented the tumor stage as a factor, and have also updated our methods accordingly (p12, r492-497). The new analysis is shown in Figure 4, suppl. Table 12, 13, and also impacted the results (p7, 249-286).

Comment: 2. Line 451, it should be “multivariable” rather than “multivariate”. This comment applies to the whole paper.

Response: Thank you. We have changed this accordingly throughout the manuscript.

Comment: 3. Figure 4 can be confusing.

3.1. I assume that one model is fitted for categorical prediction with covariates and a separate model is fitted for regression prediction with covariates. If my understanding is correct then the covariates should have different coefficient but the covariates are reported only once on each sub-figure.

3.2. Given a feature, for example, Prolif in Figure 4d, it is difficult to know which variable is more important since they have similar p-value. Given one is categorical variable and the other is continuous, the coefficients are not comparable.

Response: Thank you very much for your insightful comments. Based on your remarks, we have redone our entire survival analysis. Rather than taking the categorical predictions from classification, we are now using the continuous predictions from the classification models, i.e. the logits [0,1]. These are, for context, the same type of values that were used to plot the distribution in Figure

2C-D, or to calculate the AUROCs. As a result, we are now comparing a continuous prediction from classification with a continuous prediction from regression, enabling the direct comparison of the coefficients resulting from the Cox analysis across the tested modeling approaches. In addition, we have plotted the covariates separately to make the individual analysis for regression and classification clearer. These changes are reflected in Figure 4, suppl. Table 11, 12, 13, the corresponding numbers in the results section (p7, 249-286), and the methods (p12, r492-497).

REVIEWER COMMENTS

Reviewer #3 (Remarks to the Author):

General comments:

I recognize and appreciate the effort that the study team put in to improve the statistical content in this manuscript.

1. It is possible that we have a different definition of the word “validation”. For example, line 103, “We validated the models on CPTAC”; to me, that imply the full-retrain model as described in Supplemental Figure 6. And I would not call the full-retain model in supplemental figure 6 “retrain” at all. I recognize that different fields may use the same terminology with different meanings. It may be worthwhile to define the terminology so readers from different backgrounds can have the same understanding.
2. For Pearson’s correlation, did the authors use 5 predictions from the cross-validation as independent samples? Similar concerns were raised during the last round of review. The p-value of E-50 just looks suspicious. Please keep in mind that the 5 predictions from the cross-validation are NOT independent samples and should not be treated as independent.

Specific comments:

1. Line 119, a non-significant p-value demonstrates no statistically significant difference which does not imply equality. Please change the phrase from “all models demonstrated statistically equal AUROCs” to “there are no statistically significant differences noted in AUROCs”.
2. Line 197, similar comment as the previous one. No statistically significant difference does not imply equality. Please modify other instances of this miss-use of "equality" in the manuscript.

Reviewer #4 (Remarks to the Author): Replaces Reviewer #1; expert in cancer digital pathology, artificial intelligence and deep learning

The authors have investigated the interesting hypothesis that regression-based deep learning would outperform classification-based counterparts. They have done so with experiments using state-of-the-art computational pathology methods applied to a wide set of tasks and data sets, showing improved results from many aspects. I agree with previous reviewers that there is a good scientific contribution from this work.

In this revision, the authors have diligently addressed the reviewer comments, and substantially expanded the analysis and discussion. Not being a statistics expert, I am not able to confirm that the reworked statistical analysis is correct - although it is clear that the authors have put much effort into it.

My main comment and suggestion for revision is the use of "interpretability", primarily in the section starting line 213. I would say that interpretability is about how well the model can convey the basis for its results. (See, e.g., Graziani et al., A global taxonomy of interpretable AI: unifying the terminology for the technical and social sciences.) I would say that what is tested here is rather "the correspondence with regions of known clinical relevance". I would argue that the attention maps of the worse-performing methods could potentially provide equally good depiction of the basis of their (worse) predictions, thus having equal interpretability capacity. The experiment is nonetheless relevant, but I suggest rephrasing it to refer to the correspondence to clinical knowledge.

Another comment for the authors to look into concerns the median separation distance (lines 142-146 & 503 & suppl table 3). As I see it, the improvement measures reported could be misleading in two ways. First, the min-max normalization would be heavily influenced by an outlier being an extreme value, causing relative differences to be small for other values. Second, taking the mean of the separation distances can result in an outlier having unreasonable impact, which is the case for the 4513% improvement for TCGC-CRC. I don't think that the mean of 712% well represents the TCGA overall result, or the qualitative understanding of how much improvement CAMIL brings. For the second issues, a perhaps better alternative would be to report the absolute improvement rather than the relative one.

Finally, lines 122-123: These statements use, in my view, too strong wording. The proposed method does come out on top in most comparisons, but not for all, and the differences are often small. Thus I would mitigate "strongly suggest" and "compelling evidence" to some slightly less bold wording.

With a minor revision addressing the above concerns, I would definitely recommend acceptance of this paper.

Reviewer #5 (Remarks to the Author): Replaces Reviewer #2; expert in cancer clinical genetics and genomics, colorectal cancer, HRD and mismatch repair deficiencies

In the original submission, questions were raised about how TCGA dataset for somatic/germline results affected the results and how the regression model performed for markers like tumor mutation burden, with an eye towards eventual utility of such a method for replacing tumor/germline testing. Also,

regarding for external validation cohorts what were predictions regarding therapy response (e.g. PARP-inhibitors).

In this revision, the authors added new analysis of HRD predictions with BRCA1/2 germline/somatic mutations in breast cancer and MSI status and TMB status in colorectal cancer (Suppl. Figure 6).

Regarding therapy response predictions, this treatment data does not exist for these datasets.

Thus, for the dataset scope of this manuscript, the authors have been responsive to the concerns raised.

REVIEWER COMMENTS

Reviewer #3 (Remarks to the Author):

Comment: I recognize and appreciate the effort that the study team put in to improve the statistical content in this manuscript.

Response: Thank you very much for your positive feedback regarding the improvements of our statistical content and the acknowledgement of our effort. We have addressed your remaining comments in the responses below.

Comment: It is possible that we have a different definition of the word “validation”. For example, line 103, “We validated the models on CPTAC”; to me, that imply the full-retrain model as described in Supplemental Figure 6. And I would not call the full-retain model in supplemental figure 6 “retrain” at all. I recognize that different fields may use the same terminology with different meanings. It may be worthwhile to define the terminology so readers from different backgrounds can have the same understanding.

Response: Thank you for your valuable feedback regarding the terminology. With “validated” we mean that we take a model which was trained on cohort A and apply them on cohort B. In this specific case, we have trained on cohort A, TCGA, resulting in 5 models, which we apply to cohort B, CPTAC, to receive 5 prediction scores for each patient. These 5 prediction scores are then aggregated into 1 prediction score per patient, on which further statistical tests are performed. The terms and definitions in our study are frequently used within the realm of computational pathology and were chosen with the intention of alignment with closely related fields of application. To further clarify our terminology, we have introduced a new paragraph in our methods section “Experimental setup and implementation details” (p11-12, r508-529), which explicitly outlines our definitions of these terms.

Comment: For Pearson’s correlation, did the authors use 5 predictions from the cross-validation as independent samples? Similar concerns were raised during the last round of review. The p-value of E-50 just looks suspicious. Please keep in mind that the 5 predictions from the cross-validation are NOT independent samples and should not be treated as independent.

Response: Thank you for your comment and diligent observation of the reported p-values for the Pearson’s r. We want to clarify that we did not treat the 5 predictions as independent samples. Instead, we computed the median prediction score from the 5 models for each patient, essentially treating these 5 models as an ensemble. Consequently, every patient has only one aggregated score, and it is this aggregated single score per patient that we used for computing the Pearson’s r. The p-values resulting in E-50 were from TCGA-BRCA, which is the largest cohort for HRD (compared to the other cancer types). The extremely small p-value is presumed to be caused by the 1) strong correlation between the predictions and ground-truth and 2) the notably larger sample size of TCGA-BRCA (relative to the other cohorts) with which the metrics were calculated. The p-value for the Pearson’s r in the

software packages is calculated using the t-statistic = $\frac{r\sqrt{DoF}}{\sqrt{1-r^2}}$, with the Pearson's $r = r$ and the degrees of freedom $DoF = n_{samples} - 2$. Following the formula, a larger number of samples leads to a larger t-statistic, which typically leads to a smaller p-value down the line. Moreover, we conducted a thorough reevaluation of the p-values using a different software tool, opting for R instead of Python, to eliminate any potential software-related issues. We revised the statistics in R, specifically focusing on aggregating median prediction scores from the 5 folds. Similarly, this median aggregation resulted in one prediction per patient, where each patient prediction score is considered an independent sample in subsequent Pearson's r calculation. This reanalysis yielded exactly the same numerical outcomes (Pearson's r , p-value) as previously presented in Suppl. Tables 4 and 6. We have added the number of samples and clarification of the median aggregation method to Suppl. Table 4 and 6, accompanied by extra clarifications in the methods section (p11-12, r508-529; p12, r556-558). We have changed the notation of the p-values for the Pearson's r to avoid confusion for the reader, now reporting the extremely low p-values as $p \leq 0.0001$ in Suppl. Table 4 and 6 and in the supplementary text (p28, r937).

Comment: Line 119, a non-significant p-value demonstrates no statistically significant difference which does not imply equality. Please change the phrase from “all models demonstrated statistically equal AUROCs” to “there are no statistically significant differences noted in AUROCs”. Line 197, similar comment as the previous one. No statistically significant difference does not imply equality. Please modify other instances of this miss-use of "equality" in the manuscript.

Response: Thank you for pointing out this ambiguity in our statistical terminology. We have changed the faulty usage of statistical “equality” throughout the entire manuscript (p4, r159-160; p6, r239) and adopted your suggestion of stating there are no statistically significant differences noted, instead.

Reviewer #4 (Remarks to the Author): Replaces Reviewer #1; expert in cancer digital pathology, artificial intelligence and deep learning

Comment: The authors have investigated the interesting hypothesis that regression-based deep learning would outperform classification-based counterparts. They have done so with experiments using state-of-the-art computational pathology methods applied to a wide set of tasks and data sets, showing improved results from many aspects. I agree with previous reviewers that there is a good scientific contribution from this work. In this revision, the authors have diligently addressed the reviewer comments, and substantially expanded the analysis and discussion. Not being a statistics expert, I am not able to confirm that the reworked statistical analysis is correct - although it is clear that the authors have put much effort into it.

Response: Thank you very much for the positive assessment of our work regarding using state-of-the-art methods, our scientific contribution and our efforts towards improving our statistical methods in the revision. We have addressed your comments in the responses below.

Comment: My main comment and suggestion for revision is the use of "interpretability", primarily in the section starting line 213. I would say that interpretability is about how well the model can convey the basis for its results. (See, e.g., Graziani et al., A global taxonomy of interpretable AI: unifying the terminology for the technical and social sciences.) I would say that what is tested here is rather "the correspondence with regions of known clinical relevance". I would argue that the attention maps of the worse-performing methods could potentially provide equally good depiction of the basis of their (worse) predictions, thus having equal interpretability capacity. The experiment is nonetheless relevant, but I suggest rephrasing it to refer to the correspondence to clinical knowledge.

Response: Thank you for your feedback regarding our used terminology on interpretability. We fully agree with your remark regarding the interpretability capacity across the models - this especially holds true because the attention mechanism enabling interpretability is the same algorithm across the tested approaches. Therefore, we have rephrased our mentions on the models' interpretability throughout the manuscript to focus specifically on their ability to correspond their predictions to tissue regions of known clinical relevance (p2, r49; p3, r118-119; p6, r256-258; p6-7, r276-283; p8, r370-371; p11, r484-485). Moreover, we have added the definition of "interpretability" used in this study based on your suggestions, as well as citing the suggested paper by our co-author Mara Graziani (p12, r525-529).

Comment: Another comment for the authors to look into concerns the median separation distance (lines 142-146 & 503 & suppl table 3). As I see it, the improvement measures reported could be misleading in two ways. First, the min-max normalization would be heavily influenced by an outlier being an extreme value, causing relative differences to be small for other values.

Response: Thank you very much for your insights on min-max normalization's impact on calculating the median separation distance. While we acknowledge the effect of extreme-value outliers on min-max normalization, comparing approaches operating at

different scales necessitates a uniform measurement scale. To address this, we've now utilized a modified min-max normalization with robust scaling. This variant maintains a consistent range [0,1] for cross-approach comparison, employing min-max normalization on the 95% of data falling between the 2.5th and 97.5th percentiles of predictions for each approach. By excluding extreme values, we focus on the central data distribution, minimizing the influence of outliers on scaling. The distributions resulting from robust min-max normalization method aligns with our initial findings. Nonetheless, this adjustment enhances the applicability of our work across diverse applications. We have changed Figure 2, Suppl. Table 3, and the reporting of these metrics and methods throughout the manuscript using the aforementioned robust min-max normalization method (p5, r183-187).

Comment: Second, taking the mean of the separation distances can result in an outlier having unreasonable impact, which is the case for the 4513% improvement for TCGC-CRC. I don't think that the mean of 712% well represents the TCGA overall result, or the qualitative understanding of how much improvement CAMIL brings. For the second issues, a perhaps better alternative would be to report the absolute improvement rather than the relative one.

Response: Thank you for your comment. We acknowledge that this way of reporting results is not very intuitive. As suggested, we have now reported the absolute improvement instead, using robust min-max normalized prediction scores as described in the previous response. The new results are consistent with our previously reported findings, albeit expressed using the suggested alternative metric. Consequently, we have changed Suppl. Table 3 and the reporting of these metrics in the manuscript (p5, r183-187).

Comment: Finally, lines 122-123: There statements use, in my view, too strong wording. The proposed method does come out on top in most comparisons, but not for all, and the differences are often small. Thus I would mitigate "strongly suggest" and "compelling evidence" to some slightly less bold wording.

Response: We appreciate your feedback on the wording used in the manuscript. While our proposed method does perform favorably in most comparisons, we acknowledge that the differences can indeed be subtle. As per your suggestion, we have adjusted the wording in the mentioned statements to be less assertive, removing "strongly" and "compelling" (p4, r162-166).

Comment: With a minor revision addressing the above concerns, I would definitely recommend acceptance of this paper.

Response: Thank you very much for your positive feedback and your recommendation for acceptance of our paper.

Reviewer #5 (Remarks to the Author): Replaces Reviewer #2; expert in cancer clinical genetics and genomics, colorectal cancer, HRD and mismatch repair deficiencies

Comment: In the original submission, questions were raised about how TCGA dataset for somatic/germline results affected the results and how the regression model performed for markers like tumor mutation burden, with an eye towards eventual utility of such a method for replacing tumor/germline testing. Also, regarding for external validation cohorts what were predictions regarding therapy response (e.g. PARP-inhibitors). In this revision, the authors added new analysis of HRD predictions with BRCA1/2 germline/somatic mutations in breast cancer and MSI status and TMB status in colorectal cancer (Suppl. Figure 6). Regarding therapy response predictions, this treatment data does not exist for these datasets. Thus, for the dataset scope of this manuscript, the authors have been responsive to the concerns raised.

Response: Thank you very much for the positive assessment of our revised manuscript.

REVIEWERS' COMMENTS

Reviewer #3 (Remarks to the Author):

All my comments have been address to my satisfaction. Thank you very much.

Reviewer #4 (Remarks to the Author):

The authors have diligently and satisfactorily addressed the concerns I raised, and I am therefore happy to recommend acceptance.